# Epidemiological Study of Autoimmune Bullous Dermatoses in Northeastern Romania

**DOI:** 10.3390/diagnostics14010057

**Published:** 2023-12-26

**Authors:** Ioana Adriana Popescu, Dan Vata, Doinita Temelie Olinici, Bogdan Marian Tarcau, Adriana Ionela Patrascu, Ioana Alina Halip, Dumitrita Lenuta Gugulus, Madalina Mocanu, Laura Gheuca Solovastru

**Affiliations:** 1“Saint Spiridon” County Emergency Clinical Hospital, 700111 Iasi, Romania; ioana-adriana.popescu@umfiasi.ro (I.A.P.); doinita.p.olinici@umfiasi.ro (D.T.O.); adriana.patrascu@umfiasi.ro (A.I.P.); ioana-alina.grajdeanu@umfiasi.ro (I.A.H.); nichitean.dumi@yahoo.com (D.L.G.); madalina.mocanu@umfiasi.ro (M.M.); solovastru.gheuca@umfiasi.ro (L.G.S.); 2“Grigore T. Popa” University of Medicine and Pharmacy, 700115 Iasi, Romania

**Keywords:** autoimmune bullous dermatoses, epidemiology, Northeastern Romania

## Abstract

Background: Autoimmune bullous diseases (ABDs) are a rare but significant group of dermatoses that pose great challenges to the treating dermatologist. ABDs are characterized by the presence of tissue-bound and circulating autoantibodies directed against disease-specific target antigens of the skin. Most epidemiological studies have focused on a single ABD. More than that, there are few data about the incidence and prevalence of autoimmune blistering diseases in Romania. Methods: In this study, between 2015 and 2019, we retrospectively investigated a total of 225 patients with autoimmune bullous diseases from the northeastern region of Romania. The diagnosis was based on the clinical and histo- and immunohistological findings. Results: Pemphigus was the most frequently encountered ABD, with an incidence of 8.16/1,000,000 inhabitants, representing 58.7% (132 cases), followed by 24% cases of bullous pemphigoid (54 cases), 15.4% of patients were diagnosed with dermatitis herpetiformis (37 cases), and 0.9% other subepidermal autoimmune bullous dermatoses. The average age of onset of pemphigus vulgaris was 59.4 years, the majority of patients being male, while the average age of patients diagnosed with bullous pemphigoid was 73.8 years, the majority being female. Conclusions: Pemphigus vulgaris is the most frequently encountered ABDs in the northeast of Romania, with a higher incidence than in Western European countries, and this may be due to specific peculiarities of the geographical area, as well as to a genetic susceptibility of the population in this region.

## 1. Introduction

Autoimmune bullous dermatoses are a group of rare but debilitating skin disorders characterized by the formation of blisters and erosions on the skin and mucous membranes. These conditions are the result of the body’s immune system turning against itself, attacking proteins that are crucial for maintaining the integrity of the skin. The target proteins in ABDs are components of the desmosomes, hemidesmosomes, or basement membrane zone. Pemphigus vulgaris (PV) is primarily associated with autoantibodies targeting Desmoglein3, a desmosomal cadherin found in the mucous membranes, especially in the oral mucosa and the skin. In some cases, autoantibodies against Desmoglein1 are also present, particularly in pemphigus vulgaris with mucocutaneous involvement. The major antigen in bullous pemphigoid (BP) is BP180, also known as collagen XVII, which is a transmembrane glycoprotein present in the hemidesmosomes of the basement membrane zone. Another hemidesmosomal protein, BP230, is also targeted in some cases of BP. Understanding the specific target proteins in each ABD is crucial for accurate diagnosis and the development of targeted therapeutic strategies [1]. The name “bullous” refers to the large blisters (bullae) that form as a result of these autoimmune attacks. While these conditions are relatively uncommon, they can have a significant impact on a person’s quality of life [2]. ABDs can cause severe itching, pain, and discomfort due to blister formation and skin erosions. These physical symptoms can lead to sleep disturbances and a reduced ability to engage in daily activities. ABDs are typically chronic, with periodic flare-ups and remissions. This constant cycle of symptoms can be emotionally draining and challenging for patients to manage. The quality of life of patients with autoimmune bullous dermatoses is often significantly impacted due to the chronic and sometimes debilitating nature of these skin conditions. Assessing the quality of life of individuals with autoimmune bullous dermatoses is an important aspect of understanding the impact of these conditions on daily functioning, mental well-being, and overall health. Various standardized tools and questionnaires are used to measure the quality of life in patients with dermatological conditions, including ABDs. Some commonly used instruments include the Dermatology Life Quality Index (DLQI) or the Hospital Anxiety and Depression Scale (HADS). The DLQI is a widely used questionnaire to assess the impact of skin diseases on a person’s quality of life. It covers various aspects such as symptoms and feelings, daily activities, leisure, work and school, personal relationships, and treatment [3].

The physiopathological mechanism results from autoimmunity against intercellular adhesion molecules (like desmoglein1, desmoglein 3 in PV) or components of the basement membrane in the skin and mucosal surfaces. In ABDs, the immune system produces autoantibodies, typically immunoglobulin G antibodies, against specific proteins in the skin. These autoantibodies target structural proteins that help hold the layers of the skin together, such as desmogleins (in pemphigus) or hemidesmosomal proteins (in pemphigoid diseases). Autoantibodies interfere with the normal adhesion of skin cells, which leads to the separation of the epidermis (top layer of skin) from the dermis (deeper layer). This loss of adhesion results in the formation of blisters and erosions. The presence of activated immune cells and the release of proinflammatory cytokines like tumor necrosis factor-alpha, interleukin-1, interleukin-6, 7, 23 or interferon-gamma, contribute to inflammation in the affected skin, which can cause pain, redness, and further damage. Genetic predisposition may play a role in the development of ABDs. Certain genes related to immune regulation and skin structure can increase susceptibility to these conditions. Human leukocyte antigen (HLA) genes, particularly HLA-DR and HLA-DQ, are associated with susceptibility to pemphigus vulgaris. Different ethnic groups may show variations in HLA associations. Similar to pemphigus vulgaris, HLA class II genes are associated with bullous pemphigoid susceptibility. HLA-DQB1*03:01 has been identified as a risk allele in multiple populations [4]. In some cases, autoimmune bullous dermatoses may be triggered or exacerbated by environmental factors, such as medications, infections, or exposure to UV radiation. For example, there is evidence to suggest that the use of dipeptidyl peptidase-4 (DPP-4) inhibitors, a class of medications commonly prescribed for the management of diabetes, may be associated with an increased risk of BP. Several studies have reported an association between DPP-4 inhibitors and the development of bullous pemphigoid. DPP-4 inhibitors, such as sitagliptin, saxagliptin, linagliptin, and alogliptin, are commonly used to lower blood glucose levels in people with type 2 diabetes. The exact mechanism by which DPP-4 inhibitors may contribute to the development of BP is not fully understood, but it is thought to involve immune dysregulation. The frequency with which environmental factors contribute to ABDs is challenging to quantify precisely. ABDs are relatively rare, and the interplay between genetics and the environment is complex. Further research is needed to better understand the specific environmental triggers and their mechanisms in the context of autoimmune bullous dermatoses [5]. Over time, the immune response can broaden to target additional proteins in the skin, a phenomenon known as epitope spreading. This can lead to the evolution of the disease and the emergence of new clinical features. In PV, the initial target is typically desmoglein3, a desmosomal cadherin. However, epitope spreading can occur, leading to the recognition of desmoglein1 as well. This is particularly relevant in mucocutaneous forms of PV, where both desmoglein3 and desmoglein1 may be targeted by autoantibodies. In BP, the primary target is BP180 (collagen XVII), a protein in the hemidesmosomes of the basement membrane. Epitope spreading may involve the recognition of additional epitopes on BP180 or the targeting of another hemidesmosomal protein, BP230. The frequency and specific patterns of epitope spreading can vary among individuals and may be influenced by genetic factors, environmental triggers, and the course of the disease. Epitope spreading can have clinical implications, potentially affecting disease severity, treatment response, and the overall course of ABDs [6].

Pemphigus (PV) and bullous pemphigoid (BP) are the most prevalent autoimmune bullous dermatoses. Pemphigus can be divided into two major forms, based on the level of the blister in the epidermis. The superficial forms of pemphigus are grouped under pemphigus foliaceus, the deep forms under pemphigus vulgaris (mucosal-dominant pemphigus vulgaris and mucocutaneous pemphigus vulgaris) and its variant of pemphigus vegetants [7]. In pemphigus, IgG autoantibodies are directed against desmogleins 1 and 3, which are part of the cadherin family of cell-cell adhesion molecules. These structures are responsible for maintaining intercellular adherence in stratified squamous epithelia, such as the skin and oral mucosa. From a clinical point of view, PV initially occurs with lesions on the oral mucosa, preceding skin lesions for several months (Figure 1A,B). At this level, erosions with a small tendency to spontaneous, painful healing can be observed. The lesions start as blisters that rupture easily, causing erosions and ulcers, which can lead to serious life-threatening infections and metabolic abnormalities. A defining feature of pemphigus vulgaris is acantholysis, which refers to the detachment of keratinocytes from each other. Acantholytic cells lose their intercellular adhesion, resulting in the formation of intraepidermal blisters (Figure 1C).

Pemphigus is pretty rare and its incidence has been estimated to be about 0.2 cases per 100,000 per year in Central Europe. Bulgaria, Greece, and the Mediterranean region of Turkey have a high incidence of pemphigus [8]. The mean age of onset of the pathology is approximately 40–50 years of age, with female predominance. The pemphigoid group represents a group of autoimmune disorders characterized by subepidermal blistering. This group includes mainly BP, linear IgA disease, dermatitis herpetiformis (DH), and epidermolysis bullosa acquisita. BP most frequently affects the elderly, with the onset of disease after the age of 70 years. It leads to the development of large, tense blisters on the skin and mucous membranes (Figure 2A). Bullous pemphigoid is characterized by a subepidermal blister, meaning that the blister forms beneath the epidermal layer of the skin (Figure 2B). In this condition, the immune system targets two proteins, BP180 and BP230, which are responsible for anchoring the skin’s top and bottom layers together. Incidences are from 1.21 to 2.17 per 100,000 persons per year. The incidence of BP in Europe has more than doubled in the last decade, which might be related to the increasing age of the general population, multidrug use, and the quality of diagnosis [9].

Mucous membrane pemphigoid is the collective term for pemphigoid diseases affecting mucous membranes. The autoimmune disease targets components of the epidermal basement membrane zone. Generally, in mucous membrane pemphigoid, the oral mucosa is mostly affected, but all mucous membranes can be involved. Dermatitis herpetiformis is the skin manifestation of coeliac disease. Nowadays it is known that the trigger of both diseases is the ingestion of gluten in certain HLA phenotypes (HLA-DQ8 or HLA-DQ2) leading to an autoimmune reaction characterized by IgA autoantibodies against tissue transglutaminase and later in DH patients against the epidermal transglutaminase. DH has a different prevalence, due to its genetic background. DH is most common in patients of north-European origin. Studies report a prevalence of 1.2 to 39.2 per 100,000 persons, and an incidence range of 0.4 to 2.6 per 100,000 people per year. The onset of the disease is variable, mostly in 40 years old, but childhood and geriatric cases are not rare [10].

ABDs are chronic diseases and are associated with significant morbidity and even mortality. There is still not much data about the epidemiology of these diseases. ABDs are reported from all around the world. It is important to note that autoimmune bullous diseases are often underdiagnosed or misdiagnosed, and obtaining accurate epidemiological data can be challenging due to the rarity of these conditions. Furthermore, improved awareness among healthcare providers and standardized diagnostic criteria can impact the reported incidence and prevalence rates. Epidemiologic data on autoimmune blistering diseases contributed to the characterization of endemic and nonendemic forms of diseases, of the genetic susceptibility and environmental factors for disease etiopathogenesis. However, epidemiological data on autoimmune bullous diseases in Romania are still lacking. Therefore, this study aimed to evaluate the incidence of autoimmune bullous diseases in the northeastern region of Romania.

## 2. Materials and Methods

We performed a retrospective study on the patients with autoimmune bullous dermatoses in evidence of the clinical hospitals from Northeastern Romania, mostly from Clinical Hospital “St. Spiridon” Iasi (86% from Clinical Hospital “St. Spiridon” Iasi, 14% of patients from counties such as Neamt, Suceava or Bacau). Two hundred and twenty-five patients with autoimmune bullous diseases residing in the cities of the northeast region of Romania were enrolled in the present retrospective study between January 2014 and December 2019. Data were collected from the electronic database of the Clinical Hospitals. In this analysis, inclusion criteria were a diagnosis of pemphigus vulgaris, bullous pemphigoid, or dermatitis herpetiformis confirmed by histopathology or/and direct immunofluorescence (DIF). We mention the fact that during the study there were no cases of paraneoplastic pemphigus, pemphigoid gestationis, linear IgA disease, or epidermolysis bullosa acquisita. Exclusion criteria include a diagnosis of autoimmune bullous disease before 2014. The limitation of the study consists in the fact that only data on diagnosis, sex, and place of origin were available. The study protocol was approved by the Ethics Committee of the University of Medicine and Pharmacy in Iasi. According to the National Institute of Statistics (http://www.insse.ro), the population of the northeastern region of Romania consisted of 3,231,481 inhabitants.

Statistical analysis was performed using the SPSS version 29.0 software package (SPSS Inc., Chicago, IL, USA). Categorical variables were presented as frequencies and percentages, and continuous variables as mean ± standard deviation. Categorical and ordinal variables were compared between groups using the χ^2^ test and continuous variables were compared between groups using the Kruskal–Wallis test (when the pre-condition of normal repartition of values was not verified). Statistical significance was assessed at a value of *p* < 0.05; *p*-values < 0.01 were assessed as statistically highly significant.

## 3. Results

A total of 225 patients were included in this study based on the positive diagnosis of autoimmune bullous disease. PV was the most common diagnosis representing 132 cases (58.6%). A total of 93 patients (41.4%) were diagnosed with different sub-epidermal autoimmune blistering diseases. The incidence of PV was 8.16/1,000,000 inhabitants (Table 1-(1)).

The most frequently encountered cases were those of PV and BP, with only two cases of mucous membrane pemphigoid reported in 2015. Most cases of PV were observed in 2015 (31.8%) and 2019 (27.3%); similarly, the most frequent cases of bullous pemphigoid were also observed in 2015 (38.9%), but in 2016 (22.2%) and in 2019 (20.4%); instead, the most frequent cases of DH were reported in the years 2016 (35.1%) and 2017 (27.0%). Moreover, the most numerous observations at the level of the entire group of patients were also made at the level of 2015 (30.2%) and 2019 (24.0%), followed by 2016 (21.3%). The reduced number of patients in 2017 and 2018 can not be explained; it was probably a lower addressability. The calculated incidence for the cases of bullous pemphigoid was 3.41/1,000,000 inhabitants and for herpetiformis dermatitis 2.29/1,000,000 inhabitants.

There are no statistically significant differences between the distributions of the investigated diagnoses by counties of origin (*p* = 0.108)—most cases coming, in all situations, from Iași County—although one of the two patients with mucous membrane pemphigoid comes from Neamț County, 20.4% of those with bullous pemphigoid come from Neamț and Vaslui counties, as well as 15.9% of those with PV (Table 2).

However, no statistically significant differences can be found in terms of the distribution of patients by area of origin (*p* = 0.706) (Table 3). At the level of the entire group, most patients come from the urban area (53.8%), a phenomenon that is also noticeable separately, in patients with PV (53.0%) and those with DH (62.2%); patients with bullous or mucous membrane pemphigoid are equally distributed in the two areas of origin.

However, statistically significant differences are observed between the sexes in terms of the distribution of registered cases (*p* = 0.014) (Table 4). At the level of the entire group, a slight prevalence of the male gender (54.2%) compared to the female gender (45.8%) is observed; PV patients are almost equally distributed between the sexes, cases of bullous pemphigoid are slightly more common in women (53.7%) than in men (46.3%), while both patients with mucous membrane pemphigoid are male, at as did three-quarters (75.7%) of those with DH. Dermatitis herpetiformis affects both males and females, but there may be a slight male predominance in some populations, as in our study, but with a high variance in favor of the male sex.

Instead, statistically significant differences are found between the four diagnostic categories investigated in terms of the average age of onset and age category respectively (*p* < 0.001) (Table 5 and Table 6). In general, all investigated patients are elderly, with an average age of 61.88 ± 16.617 years; the oldest are the two patients with mucous membrane pemphigoid, with an average age of 82.50 ± 12.021 years, and those with bullous pemphigoid, with an average age of 73.85 ± 12.410 years; on the other hand, the youngest are the DH patients, with an average age of 51.81 ± 20.884 years.

Most of the investigated patients are between 51 and 80 years old (69.4%) and almost one-third (28.0%) are between 51 and 60 years old. Most cases of PV (36.4%) were noted in this age range (51–60 years), while most cases of bullous pemphigoid (68.5%) as well as both patients with mucous membrane pemphigoid are aged over 70 years. Most cases with DH were also observed between 51 and 60 years (24.3%), but also under 40 years (21.6%).

The average age of onset of PV was 59.4 years, the majority of patients being male, while the average age of patients diagnosed with bullous pemphigoid was 73.8 years, the majority being female.

## 4. Discussion

In Romania, there are few epidemiological data about autoimmune bullous dermatoses. Even if they are considered rare pathologies, it seems that their incidence is constantly increasing. In this sense, we have analyzed the incidence of autoimmune bullous dermatoses in the northeast of Romania, corroborating data from regional hospitals related to our region. The proportions of the centers involved are very unequal and make it difficult to draw general conclusions about regional events, most likely because most patients go to university hospitals, such as the one in Iasi, Romania. We discovered that PV and not the pemphigoid group was the most common autoimmune pathology in Northeastern Romania, in terms of incidence (0.816/100,000 inhabitants). Compared to epidemiological studies from other regions of the world, the incidence of PV in Romania is similar to that in Greece or Tunisia, higher than that in Western Europe, and lower than the incidence in Israel [11,12]. Multiple studies suggest that PV displays a heterogeneous geographic and ethnic distribution. The estimated annual incidence rate of PV is typically reported to be around 0.1 to 2.4 cases per 100,000 individuals. The incidence of PV may show geographical variation (Appendix A). It has been reported to be more prevalent in certain ethnic groups, including individuals of Jewish descent and those with Mediterranean, Middle Eastern, or South Asian ancestry [13,14]. PV affects both males and females, but some studies suggest a slightly higher prevalence in females. PV can occur at any age but most commonly presents in middle-aged or older adults. The peak age of onset is often between 40 and 60 years. In our study, the average age of onset of PV was 59.4 years and the majority of patients were male.

The incidence of BP in Europe has more than doubled in the last decade, which might be related to the increasing age of the general population, multidrug use, and the quality of diagnosis [4]. BP most frequently affects the elderly, with disease onset after the age of 70 years. Incidences are from 1.21 to 2.17 per 100,000 persons per year. Interestingly, the incidence of bullous pemphigoid (0.341/100,000 inhabitants) in the northeast of Moldova is in the opposite direction, with a lower value than in Western Europe, but close to the incidence calculated in the northwestern region of Romania [14]. The genetic background of the studied populations may matter a lot for this difference. Also, other factors could be involved in the results of the epidemiological data. Bullous pemphigoid is a pathology found in elderly people, in the studied group with an average age of 73. As is well known, life expectancy in Romania is lower, compared with the more developed countries. In this context, some patients could be underdiagnosed, especially those with mild forms of the disease. A definite diagnosis of bullous pemphigoid can be very easily overlooked, without performing paraclinical tests to confirm the diagnosis.

DH has a different prevalence, due to its genetic background [10,15]. DH is associated with specific human leucocyte antigen (HLA) class II: HLA-DQ2 and HLA-DR3 as well as class I: HLA-A1 and HLA-B8. DH is most common in patients of north-European origin. Studies report a prevalence of 1.2 to 39.2 per 100,000 persons, and an incidence range of 0.4 to 2.6 per 100,000 people per year. In our study, the incidence of DH was 0.229/100,000 inhabitants, with an average age of onset of 51 years, contrary to the existing data in the literature, in which the average age of onset is approximately 40 years. However, 21.6% of cases were observed in patients under 40 years old, and 18% in patients over 70 years old, affecting more men than women, compliant with data from other reported studies [15].

Diagnosis of autoimmune bullous dermatoses usually results from clinical features, histological examination, and the quantification of circulating typical autoantibodies. Accurate diagnosis is fundamental to successful management. Immunofluorescence studies, serological tests, and skin biopsies are essential for confirming the specific autoimmune bullous dermatosis and determining its subtype. Subtyping is crucial as different conditions may require distinct treatment approaches. The treatment of autoimmune bullous dermatoses is typically aimed at suppressing the autoimmune response, reducing inflammation, and managing symptoms. The approach may vary based on the specific type of ABD and the severity of the condition. Due to chronic evolution, their treatment and management still represent a challenge due to the higher frequency of several comorbidities in this group of patients. Therefore, an early diagnosis and prompt correct treatment are mandatory to reach better clinical outcomes and improve as much as possible the outcomes. It is important to note that the choice of treatment depends on factors such as the type and severity of the ABD, patient age, comorbidities, and individual response to therapy. Treatment plans are often tailored to the specific needs of each patient, and close monitoring is essential to assess response and adjust the approach as needed. Additionally, ongoing research may lead to the development of new and more targeted therapies.

Since systemic therapies may be necessary for a long time, the severity of the disease should be thoroughly evaluated at baseline and follow-up sessions to evaluate clinical and biological response to treatment. Regarding clinical monitoring, it is currently advised that disease severity and treatment response should be assessed using ABD-specific criteria and outcome measures.

Autoimmune bullous dermatoses are challenging conditions that can have a significant impact on a person’s physical and emotional well-being. While these disorders primarily affect the skin, their impact extends beyond the physical realm, profoundly influencing an individual’s psychic-emotional well-being. The emotional and psychological consequences of autoimmune bullous dermatoses can be substantial. Patients often experience feelings of anxiety, depression, and reduced self-esteem due to the visible and sometimes disfiguring nature of the skin lesions. Social interactions may become strained as individuals cope with the stigma and misunderstanding associated with their condition. More specific instruments are also described in the literature, such as the Autoimmune Bullous Disease Quality of Life (ABQOL) questionnaire and Treatment of Autoimmune Bullous Disease Quality of Life (TABQOL) questionnaires, used to measure ABDs’ burden and to evaluate responses to therapeutic intervention [16].

The chronic nature of autoimmune bullous dermatoses and the need for ongoing medical treatment can lead to feelings of frustration and hopelessness. Healthcare providers must recognize the psychic–emotional impact and provide not only medical treatment but also psychological support and counseling to help patients cope with the challenges posed by autoimmune bullous dermatoses. Early diagnosis and appropriate treatment are essential for managing these disorders effectively. While there is no cure for autoimmune bullous dermatoses, advances in medical research and treatment options continue to provide hope for those affected, improving their quality of life and reducing the burden of these conditions. With the support of healthcare professionals and the development of new therapies, individuals living with autoimmune bullous dermatoses can better manage their symptoms and maintain healthier skin.

In conclusion, this study is the second conducted in Romania, the first in the northeastern region, which, in addition to the epidemiological data provided, also allows the calculation of a possible increase in incidence, a phenomenon that has recently been observed in other countries.

The resulting data demonstrate a higher incidence of PV compared to BP and other subepidermal bullous dermatoses in the northeastern region of Romania. Possible causes could include a certain genetic susceptibility of the population in the northeastern region of Romania, a shorter life expectancy, or an underestimation of the true prevalence of BP but further epidemiological studies may contribute to these findings in this geographic area. Epidemiological studies can help identify specific risk factors, such as genetic predisposition, environmental triggers, or comorbid conditions, which can inform future research efforts. Understanding the underlying causes and mechanisms of ABDs is crucial for developing better treatment strategies. A better understanding of the epidemiology of ABDs can attract more research funding and pharmaceutical interest, potentially leading to the development of novel therapies and treatment options. In summary, future epidemiological studies on autoimmune bullous dermatoses are essential for enhancing our understanding of these rare skin disorders, improving patient care, and ultimately advancing the development of more effective treatments and support systems for individuals affected by ABDs.

## Figures and Tables

**Figure 1 diagnostics-14-00057-f001:**
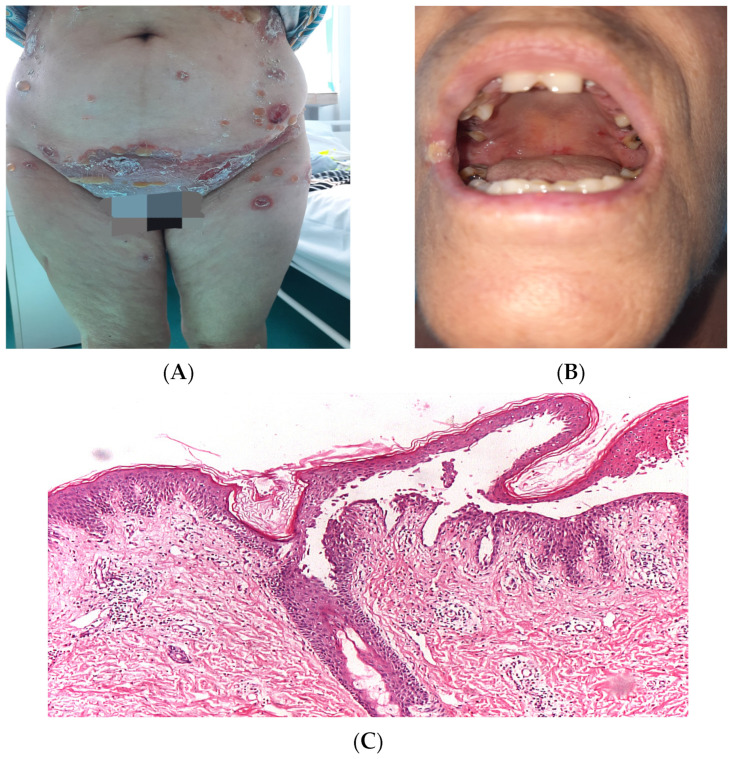
A 58-year-old woman diagnosed with pemphigus vulgaris; (**A**,**B**)—clinical aspect; (**C**)—suprabasal acantholysis histopathological aspect (HE ×5).

**Figure 2 diagnostics-14-00057-f002:**
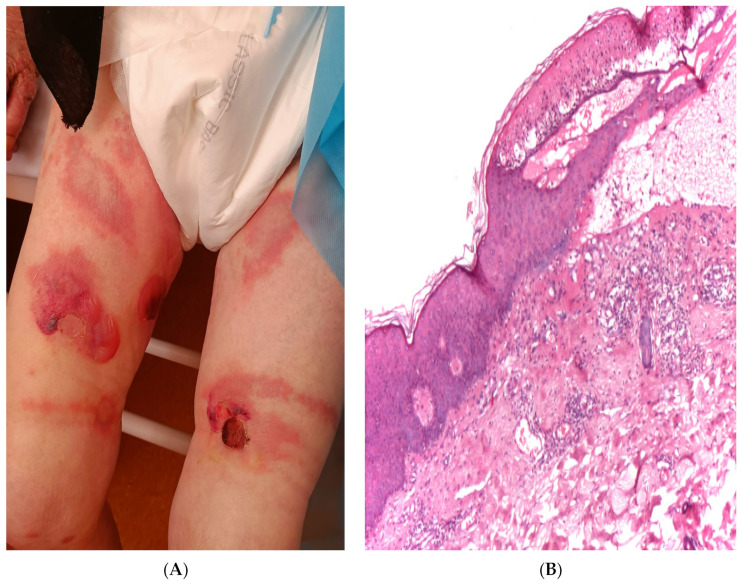
An 83-year-old woman diagnosed with bullous pemphigoid; (**A**)—clinical aspect; (**B**)—subbasal acantholysis, histopathological aspect (HE ×10).

**Table 1 diagnostics-14-00057-t001:** (1) The resident population of the northeast region and the incidence per 100,000 habitants. (2) Incidence of autoimmune blistering diseases in the studied group.

(1)											
The Resident Population of the Northeast Region						Incidence per 100,000 Habitants					
	2015	2016	2017	2018	2019	2015	2016	2017	2018	2019	
Total	3,265,089	3,248,746	3,231,764	3,213,370	3,198,435						Average
ABDs	68	48	33	22	54	2.083	1.477	1.021	0.685	1.688	1.391
PV	42	23	20	11	36	1.286	0.708	0.619	0.342	1.126	0.816
PB	21	12	3	7	11	0.683	0.369	0.093	0.218	0.344	0.341
MMP	2					0.061					0.061
DH	3	13	10	4	7	0.092	0.400	0.309	0.124	0.219	0.229
**(2)**											
		**Pemphigus**		**Bullous pemphigoid**		**Mucous membrane pemphigoid**		**Herpetiformis dermatitis**		**Total**	
		** *N* **	**%**	** *N* **	**%**	** *N* **	**%**	** *N* **	**%**	** *N* **	**%**
		**Pearson Chi-square = 25.425; *p* = 0.013**									
Year	2015	42	31.8%	21	38.9%	2	100.0%	3	8.1%	68	30.2%
	2016	23	17.4%	12	22.2%			13	35.1%	48	21.3%
	2017	20	15.2%	3	5.6%			10	27.0%	33	14.7%
	2018	11	8.3%	7	13.0%			4	10.8%	22	9.8%
	2019	36	27.3%	11	20.4%			7	18.9%	54	24.0%
Total		132	100.0%	54	100.0%	2	100.0%	37	100.0%	225	100.0%

**Table 2 diagnostics-14-00057-t002:** Distribution of cases by counties of origin.

		Pemphigus		Bullous Pemphigoid		Mucous Membrane Pemphigoid		DH		Total	
		*N*	%	*N*	%	*N*	%	*N*	%	*N*	%
		Pearson Chi-Square = 10.433; *p* = 0.108									
County	Iasi	111	84.1%	43	79.6%	1	50.0%	36	97.3%	191	84.9%
	Neamt	12	9.1%	6	11.1%	1	50.0%			19	8.4%
	Vaslui	9	6.8%	5	9.3%			1	2.7%	15	6.7%
Total		132	100.0%	54	100.0%	2	100.0%	37	100.0%	225	100.0%

**Table 3 diagnostics-14-00057-t003:** Distribution by area of origin.

		Pemphigus		Bullous Pemphigoid		Mucous Membrane Pemphigoid		DH		Total	
		*N*	%	*N*	%	*N*	%	*N*	%	*N*	%
		Pearson Chi-Square = 1.398; *p* = 0.706									
Area	Rural	62	47.0%	27	50.0%	1	50.0%	14	37.8%	104	46.2%
	Urban	70	53.0%	27	50.0%	1	50.0%	23	62.2%	121	53.8%
Total		132	100.0%	54	100.0%	2	100.0%	37	100.0%	225	100.0%

**Table 4 diagnostics-14-00057-t004:** Distribution by gender.

		Pemphigus		Bullous Pemphigoid		Mucous Membrane Pemphigoid		DH		Total	
		*N*	%	*N*	%	*N*	%	*N*	%	*N*	%
		Pearson Chi-Square = 10.554; *p* = 0.014									
Sex	Female	65	49.2%	29	53.7%			9	24.3%	103	45.8%
	Male	67	50.8%	25	46.3%	2	100.0%	28	75.7%	122	54.2%
Total		132	100.0%	54	100.0%	2	100.0%	37	100.0%	225	100.0%

**Table 5 diagnostics-14-00057-t005:** The average age of onset.

Diagnostic	*N*	Average	Std. the Error of the Average	Std. Deviation	Min	Max	Na Average
	Kruskal-Wallis H = 52.538; *p* < 0.001						
Pemphigus	132	59.48	1.196	13.743	22	86	59.00
Bullous pemphigoid	54	73.85	1.689	12.410	15	89	75.00
Mucous membrane pemphigoid	2	82.50	8.500	12.021	74	91	82.50
DH	37	51.81	3.433	20.884	7	78	60.00
Total	225	61.88	1.108	16.617	7	91	63.00

**Table 6 diagnostics-14-00057-t006:** Distribution by age category.

Age Range	Pemphigus		Bullous Pemphigoid		Mucous Membrane Pemphigoid		DH		Total	
	*N*	%	*N*	%	*N*	%	*N*	%	*N*	%
	Pearson Chi-Square = 67.526; *p* < 0.001									
≤40 years	13	9.8%	1	1.9%			8	21.6%	22	9.8%
41–50 years	16	12.1%					6	16.2%	22	9.8%
51–60 years	48	36.4%	6	11.1%			9	24.3%	63	28.0%
61–70 years	23	17.4%	10	18.5%			7	18.9%	40	17.8%
71–80 years	26	19.7%	19	35.2%	1	50.0%	7	18.9%	53	23.6%
>80 years	6	4.5%	18	33.3%	1	50.0%			25	11.1%
Total	132	100.0%	54	100.0%	2	100.0%	37	100.0%	225	100.0%

## Data Availability

The data presented in this study are available on request from the corresponding author.

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
