# Peer review of "Epidemiological Study of Autoimmune Bullous Dermatoses in Northeastern Romania"

_diagnostics, 2023, doi:10.3390/diagnostics14010057_

Round 1
Reviewer 1 Report
Comments and Suggestions for Authors
The authors reported the results of a retrospective study including a total of 15 225 patients with autoimmune bullous diseases from the Northeastern region of Romania, aiming to describe epidemiological data on autoimmune bullous diseases in Romania.
The study is overall clear and well-written.
I suggest improving the discussion section, briefly discussing the diagnosis and briefly also the management of these diseases. (Strategies to Improve Outcomes of Bullous Pemphigoid: A Comprehensive Review of Clinical Presentations, Diagnosis, and Patients' Assessment. Clin Cosmet Investig Dermatol. 2022;15:661-673.)
There are some minor typos to correct.
Comments on the Quality of English Language
minor typos to correct
Reviewer 2 Report
Comments and Suggestions for Authors
This manuscript by Popescu et al. is a retrospective epidemiologic analysis of the incidence and prevalence of autoimmune bullous dermatoses in the northeastern region of Romania. The topic of the manuscript is important to close regional data gaps and to be able to recognize trends. However, there are some comments on the manuscript:
Introduction:
Line 33: talking only about proteins is too global here. It would be good if examples were given at this point.
Lines 40-42: data e.g. on DLQI or HADS are missing here to show the severity of the influence.
Line 44: "adhesion molecules" - examples would also be necessary here
Line 52: which pro-inflammatory cytokines?
Line 54: which genes ?
Line 55: missing letter at the beginning of the sentence
Line 55: how often does an exacerbation or manifestation occur due to environmental factors?
Line 58: "additional proteins" - which ones are meant and how often does this occur?
Line 68: The abbreviation "PV" was not introduced in the text
Figure 1: the connection to this figure is missing
Line 88: Typo "multi-drog"
Figure 2: Missing reference to figure
Line 92: MMP is not the name of a patient group but a collective term for pemphigoid diseases affecting mucous membranes. Subtypes would be e.g. OMMP and LVP.
Introduction as a total: The introduction was only referenced with 5 sources. Additional sources should be included here
Material and methods:
Lines 119-121: The proportionate involvement of the various centers should already be listed here
Line 124: Time span is redundant to line 121
Line 125: Which exclusion criteria were set?
Material and methods section as a whole: The methodology should be described in more detail and the statistics should also be presented more extensively. Which statistical tests were used for which data and how were the significance levels set for the rejection of the null hypothesis?
Results:
Table 1: How can the reduced number of patients in 2017 and 2018 be explained?
Table 2 and Table 4 both refer to geographical aspects and should be combined into one table.
Table 3: it is not evident from the table which comparison the p-value refers to
Line 188: how can the high variance be explained?
Table 5: to which comparison does the p-value refer? What does the column "Na average" represent? Was a posthoc test performed, e.g. Dunn-Bonferroni test?
Table 6: How were the missing values for DH handled for the statistics?
Overall results: The numerous small tables and interspersed text sections produce a cluttered picture. It should be checked whether tables can be merged in a meaningful way. Overall, the tables contain very little information.
Discussion:
Lines 205-206: The proportions of the centers involved are very unequal and make it difficult to draw general conclusions about regional events. The imbalance should be emphasized more clearly.
Line 213: Abbreviation "PV" was not explained in the test
Lines 214-216: How is the comparison of the incidence of PV in Finland with that within the Jewish population in the USA justified? Why was the maximum of PV on the value of a specific ethnic group within a country explicitly chosen here, when the manuscript only examines the regional incidence of Romania?
I recommend inserting Table 7 in the supplement
Line 226: more reference should be made here to genetics. Which genes and what is the effect
Lines 245-247: Can this be proven using DLQI and HADS questionnaires?
Line 250: Typo
Lines 264-269: Has an increasing incidence also been reported in other dermatologic conditions during the period?
References:
The citation of literature in the text is not clear and in the list of references 37 are listed although the last citation in the discussion in line 240 ends with the number 10. The formatting in the list is not correct and several numbers are assigned to a paper, e.g. references 16 and 17
Comments on the Quality of English LanguageThe text contains spelling mistakes and missing letters and the sentence structure should also be more rigorous.
Reviewer 3 Report
Comments and Suggestions for Authors
1. Some minor English mistakes should be corrected, for example line 122 "Two hundred twenty and fifty patients", line 14 "there are few dates about the incidence"??? etc.
2. It should be helpful to add the population of the counties mentioned in paragraph 3.2
3. Every dermatologist knows it, but to be typically correct you should not use sometines "pemphigus vulgaris" and sometimes "PV". Abbreviate it everywhere after you explain it the first time you use it. And everything else of course.
4. Most reference numbers are wrong, especially those in Table 7.
5. Again in table 7, no country named Macedonia ever existed, according to UN. It is now called North Macedonia and before that FYROM.
6. The most important information that the manuscript lacks, is data about the patients. It should be helpful to the reader to know, for example, the comorbidities of the patients, or the prevalence of some common drugs. If no more data are available than sex, age and place of origin, it should be mentioned as a study limitation. By the way, there are not any limitations mentioned at all.
6.1 You analyze very well in discussion the possible explanations about your findings considering BP and PV. But the first thing a dermatologist would ask after seeing your numbers, would be the possible usage (or not) of dipeptidyl peptidase-4 inhibitors for diabetes, as we all know that, in the last years they are probably the most important cause of BP.
7.Is there any explanation as of why you did not encounter any case of paraneoplastic pemphigus, pemphigoid gestasionis, linear IgA disease etc. Maybe it is pure luck but you should mention that.
Comments on the Quality of English Language
-
Round 2
Reviewer 2 Report
Comments and Suggestions for Authors
1. Redundancy: Completely identical sentence in introduction and discussion:
Various standardized tools and questionnaires are used to measure quality of life in patients with dermatological conditions, including ABDs. Some commonly used instruments include Dermatology life quality index (DLQI) or Hospital anxiety and drepression scale (HADS) [3].
Various standardized tools and questionnaires are used to measure quality of life in patients with dermatological conditions, including ABDs. Some commonly used instruments include Dermatology life quality index (DLQI) or Hospital anxiety and drepression scale (HADS) [36].
The authors talk about a reduction in patients' quality of life, but do not substantiate this with data that can be determined using the DLQI or HADS. The mere mention of such instruments does not support the authors' statement. More specific instruments are also described in the literature, such as the Autoimmune Bullous Disease Quality of Life (ABQOL) questionnaire.
2. add magnification of H&E staining (fig 1c, 2b)
3. do the authors mean „multridrug use“ instead of „multidrog use“?
4. doubled word: „and and place of origin were available“
5. reference 11 and 12 are identically
6. reference 39 and 40 belong together
Comments on the Quality of English Languagesome typos, word doublings, missing spaces
Reviewer 3 Report
Comments and Suggestions for Authors
There are still some minor spelling mistakes but i cant point them all out because the revised manuscript doesnt have numbered lines. For example, page 4: multidrog, page 8: witj instead of with, page 10: hospitald, etc.
References are all wrong. 14-15 and 20-21 are the same article and everything has wrong number from there on.
Comments on the Quality of English Language-
